# Prenatal Versus Postnatal Diagnosis of Meckel–Gruber and Joubert Syndrome in Patients with *TMEM67* Mutations

**DOI:** 10.3390/genes12071078

**Published:** 2021-07-16

**Authors:** Agnieszka Stembalska, Małgorzata Rydzanicz, Agnieszka Pollak, Grazyna Kostrzewa, Piotr Stawinski, Mateusz Biela, Rafal Ploski, Robert Smigiel

**Affiliations:** 1Department of Genetics, Wroclaw Medical University, 50-368 Wroclaw, Poland; 2Department of Medical Genetics, Medical University of Warsaw, 02-106 Warsaw, Poland; mrydzanicz@wum.edu.pl (M.R.); Poli25@wp.pl (A.P.); g.kostrzewa@wp.pl (G.K.); stawinski84@gmail.com (P.S.); rploski@wp.pl (R.P.); 3Department of Paediatrics, Division of Paediatric Propedeutics and Rare Disorders, Wroclaw Medical University, 51-618 Wroclaw, Poland; mateuszbiela14@gmail.com

**Keywords:** *TMEM67* gene, prenatal and postnatal diagnosis, genetic and phenotypic diagnosis, Joubert syndrome, Meckel–Gruber syndrome

## Abstract

Renal cystic diseases are characterized by genetic and phenotypic heterogeneity. Congenital renal cysts can be classified as developmental disorders and are commonly diagnosed prenatally using ultrasonography and magnetic resonance imaging. Progress in molecular diagnostics and availability of exome sequencing procedures allows diagnosis of single-gene disorders in the prenatal period. Two patients with a prenatal diagnosis of polycystic kidney disease are presented in this article. *TMEM67* mutations were identified in both fetuses using a whole-exome sequencing (WES) study. In one of them, the phenotypic syndrome diagnosed prenatally was different from that diagnosed in the postnatal period.

## 1. Introduction

Renal cystic diseases are a group of malformations of the kidney that develop in fetal life or occur after birth but may not be visible until adulthood. Renal cysts can be congenital or acquired, focal, multifocal, one-sided, or bilateral [1,2,3]. There are several classifications of renal cysts [1,2,4]. Congenital renal cystic disorders are a genetically heterogeneous group and can be isolated pathologies or can be associated with other nephrological/urological disorders (ureteropelvic and ureterovesical junction obstruction, ureterocoele, posterior urethral valves, and prune-belly syndrome) as well as with systemic disorders (e.g., tuberous sclerosis, Von Hippel–Lindau syndrome, various ciliopathies) [2,3,4,5]. It is important to distinguish between genetic and non-genetic causes because of the high risk of recurrence in the case of a genetic disorder. In cases associated with extrarenal anomalies, the risk of a genetic etiology is increased, while genetic pathology is rare in the case of solitary cysts with normal renal parenchyma and in isolated unilateral multicystic dysplastic kidney or cystic dysplasia [1,3].

The prognosis of renal cysts depends on their etiology and coexisting multiple anomalies. The presence of additional defects or oligohydramnios (especially earlier-onset) may increase the risk of intrauterine or postnatal death [2,4].

We report two patients with a prenatal diagnosis of polycystic kidney disease with *TMEM67* mutations identified in a whole-exome sequencing (WES) study. In one of the presented patients, the phenotypic diagnosis made prenatally was different from that made in the postnatal period. The first proband with *TMEM67* mutations suffers from polycystic kidney and brain anomalies, which prenatally were diagnosed as Meckel–Gruber syndrome, but postnatally were clinically verified as Joubert syndrome. In the second proband with *TMEM67* mutations, Meckel–Gruber syndrome was correctly diagnosed prenatally and confirmed in an autopsy.

## 2. Clinical Reports

### 2.1. Case #1

The 32-year-old woman has been referred for genetic counseling in the second trimester of her second pregnancy due to fetal polycystic kidneys, a suspicion of agenesis of the cerebellar vermis, and heart defect (multiple muscular defects in the ventricular septum without hemodynamic significance). In prenatal screening in the first trimester, only fetal tricuspid regurgitation was observed. Screenings for fetal trisomies in the first trimester and echocardiography in the second trimester of pregnancy were normal.

The woman had a healthy son from another relationship; family history was negative. The prenatal genetic diagnosis was performed in the sixteenth week of the present pregnancy. After a normal result of karyotype and aCGH test, whole-exome sequencing (WES) was performed due to suspected Meckel–Gruber syndrome. In the echocardiography in the third trimester of pregnancy (32 weeks), multiple muscular defects were observed in the ventricular septum without hemodynamic significance.

The child was born by cesarean section at term at 38 weeks with symptoms of birth asphyxia (neonatal parameters: weight 3100 g, length 52 cm, OFC 34 cm). Respiratory failure, perinatal hypoxia, and muscular hypotonia were observed, and polycystic kidneys and ventricular septum defects were confirmed. Brain imaging (USG, CT) after birth showed no changes; cerebellar structures were described as normal. No typical facial dysmorphic features were recognized (Figure 1).

In the fifth month of life, rotatory nystagmus was observed. In addition to nystagmus, convergent strabismus and mild visual impairment have been described in an ophthalmological examination. In an electrophysiological examination of visual pathways, a bilaterally symmetrical response with normal latency was observed. In MRI of the brain, agenesis or significant hypoplasia of the cerebellar vermis with distortion of the lumen of the fourth chamber ventricle (“bat-wing fourth ventricle”) and elongation of the upper cerebellar peduncles was observed. The presence of the molar tooth sign in an MRI study changed the diagnosis from Meckel–Gruber syndrome to Joubert syndrome.

In the first year of life, developmental delay and hypertension with secondary hypertrophic cardiomyopathy were additionally observed. Despite rehabilitation, the child does not sit up alone in the fourteenth month of life. The development of speech is delayed.

### 2.2. Case #2

The 36-years old woman has been referred to the genetic clinic in week 21 of her fifth pregnancy. She has one healthy daughter. The family history was notable for the spontaneous loss of three of the mother’s prior pregnancies in the first trimester. Prenatal ultrasound screening in the first trimester of the fifth pregnancy was normal. Low risks of all trisomies were noted in biochemical screening tests. Severe oligohydramnios and later anhydramnios as well as fetal polycystic kidneys, occipital encephalocele, incorrect position of the heart, and a very small chest were observed in an ultrasound examination at 21 weeks of this pregnancy. Limbs were not visible. Amniocentesis was performed, and aCGH and afterward, WES tests were ordered. The pregnancy was terminated at 22 weeks.

## 3. Methods of Genetic Analyses

Classical cytogenetic analysis of cultured amniocytes was performed [6]. Chromosome GTG banding analysis was conducted on 15 metaphases at a 400–450 band level according to ISCN 2016 [ISCN 2016], with the use of an Imager.M1 (Zeiss) microscope and Ikaros software (Metasystems DE). To identify fetal microdeletions/microduplications, an array CGH was applied using the Agilent SurePrint G3 Human CGH Microarray Kit 4 × 180 K platform (Agilent) and Agilent CytoGenomics Edition 2.7.8.0 Software (Agilent). All diagnostic procedures were conducted according to the producer’s protocols.

Proband’s DNA isolated from amniocytes was subjected to NGS-based WES performed in the RAPID form according to a previously described protocol [7]. Venous blood samples were collected from the proband’s parents. WES was performed on the probands’ DNA using SureSelectXT Human All Exon v7 (Agilent, Agilent Technologies, Santa Clara, CA, USA) for case #1, and the Nextera Flex for Enrichment sample preparation kit (Illumina, San Diego, CA, USA) combined with TruSeq DNA Exome probes (Illumina) for case #2. Enriched libraries were paired-end sequenced (2 × 100 bp) on NovaSeq 6000 (Illumina). Variants considered as plausible causative were further validated in the probands and studied in their parents by amplicon deep sequencing (ADS) performed using Nextera XT Kit (Illumina) and paired-end sequenced (2 × 100 bp) on HiSeq 1500 (Illumina).

## 4. Results of Genetic Tests

Prenatal cytogenetic analysis revealed normal male fetus karyotype (46, XY). Array CGH showed no microduplication/microdeletion.

In both cases, the WES study revealed a compound heterozygote in *TMEM67* gene: NM_153704.5: c. [1843 T > C]; [1927 C > T]/p. (Cys615Arg); (Arg643Ter) in case #1 and NM_153704.5: c. [737G > A]; [1975C > T]/p. (Cys246Tyr); (Arg659Ter) in case #2 (Figure 2) inherited from healthy parents. Variant p. (Cys246Tyr) has zero frequency in all the tested databases, including an in-house database of >3000 Polish individuals analyzed by WES. The characteristics of the identified variants are given in Table 1 [8].

## 5. Discussion

Mutations in *TMEM67* are inherited in an autosomal recessive manner and may cause a variety of conditions such as Meckel–Gruber syndrome 3 (MKS3; OMIM 607361), COACH syndrome (OMIM 216360), Bardet–Biedl syndrome 14 (BBS14; OMIM 615991), RHYNS syndrome (OMIM 602152) as well as Joubert syndrome 6 (JBTS6; OMIM 610688) and nephronophthisis 11 (NPHP11; OMIM 613550). The *TMEM67* gene encodes the transmembrane protein 67, localizing to the primary cilium and to the plasma membrane [1]. Mutations in this gene result in dysfunction of primary cilia during early embryogenesis. Therefore, all the syndromes mentioned above belong to the group called ciliopathies [9]. Many different genes are involved in the maintenance of cilia, and the various mutations in these genes may cause a wide spectrum of symptoms and dysfunctions of many organs [1,9,10].

Diagnosis of the above syndromes is based on clinical features and genetic testing and is a difficult process in the prenatal period. These syndromes are defined as a set of features that may not all be present at the time of diagnosis, especially prenatally. It is difficult to determine the clinical criteria for their prenatal diagnosis. The diagnosis of multiple birth defects in a fetus is an indication of genetic testing, the results of which should be correlated with changing clinical symptoms observed in the following months of fetal life as well as after birth.

We present clinical reports of two fetuses with two different compound heterozygous mutations in the *TMEM67* gene recognized in a WES study performed in the rapid mode. One fetus with *TMEM67* mutations was prenatally misdiagnosed as MKS3, and the second one was correctly diagnosed prenatally as MKS3. Meckel–Gruber syndrome (MKS) is a rare disorder with multiple birth defects in the characteristic clinical triad: central nervous system abnormalities (most commonly: occipital encephalocele, hydrocephalus, anencephaly, holoprosencephaly, Dandy–Walker syndrome), and bilateral renal polycystic kidneys with cystic dysplasia and polydactyly. Other related conditions are lip/palate cleft, heart defects, genital anomalies (common cryptorchidism in males), liver fibrosis, and skeletal defects [11,12,13]. There are no clear minimum clinical criteria for the diagnosis of MKS. However, while polycystic kidneys are the most common and constant feature of this syndrome, postaxial polydactyly is the most variable feature of MKS [14,15,16]. It is suggested that two of the three classical features or one classical and two other anomalies are sufficient for a clinical diagnosis of Meckel–Gruber syndrome [15]. On this basis, MKS was clinically suspected in the presented fetuses. The first described fetus showed polycystic kidneys, cerebellar defect, and a heart defect, while polycystic kidneys and occipital encephalocele were observed in case #2. Clinical suspicions were confirmed by identifying two mutations in the *TMEM67* gene (case #1: one known pathogenic variant and another, new, potentially pathogenic variant; case #2: one potentially pathogenic variant and another, pathogenic variant, both novel). Carriership of a single mutation was identified in the parents of both fetuses.

The *TMEM67* gene is 1 of 14 known genes whose mutations are responsible for MKS. *TMEM67* mutations are responsible for 16% of cases of Meckel–Gruber syndrome [13,14,17]. Usually, MKS is lethal in utero or in the very early neonatal period [10,11,14]. According to Barisic et al., in the population consisting of 191 cases of MKS identified between January 1990 and December 2011 in 34 European registries, only a few children survived the first weeks of life [15]. Pregnancy loss in the mother of the presented case #2 may be connected with the presence of Meckel–Gruber syndrome in spontaneously aborted fetuses. Similar cases of first-trimester pregnancy loss in families with MKS were previously described [18].

In case #1, the child was born at term. He had breathing problems after birth but without fatal effect. Further observation of the child and re-evaluation of his MRI of the brain allowed the change of the diagnosis to Joubert syndrome. This syndrome belongs to a clinically and genetically heterogeneous group of disorders characterized by a broad range of symptoms. More than 30 genes causing Joubert syndrome were found, among which *TMEM67* was one of the most common genes [19,20]. The diagnosis of Joubert syndrome is commonly made after birth; however, a few cases of prenatal diagnosis were noted [12,19]. The diagnosis of Joubert syndrome is based on three clinical features: (1) the “molar tooth sign” accompanied by cerebellar vermis hypoplasia, (2) hypotonia or low muscle tone, and (3) developmental delays or intellectual disability [12]. There is a broad clinical heterogeneity in this syndrome resulting in subcategories, which may be useful in clinical practice [12,21]. Backward analysis of prenatal renal symptoms in the presented case #1 shows that they could correspond with features characteristic of Joubert syndrome and were previously described by others [20,22,23,24,25,26,27]. However, usually, nephronophthisis and cystic dysplasia spectrum were observed in the majority of cases of Joubert syndrome with *TMEM67* mutations. In the presented case #1, Joubert syndrome with renal disease (JS-Ren) was eventually diagnosed. In this form, end-stage renal disease (ESRD) occurs at a median of 13 years [12,28].

The first presented proband is under multidisciplinary care, including a pediatrician, neurologist, cardiologist, nephrologist and urologist, and ophthalmologist, as well as a physiotherapist, speech therapist, and psychologist. The family in the two presented cases has received genetic counseling. The parents of the child with Joubert syndrome or with Meckel–Gruber syndrome associated with *TMEM67* mutations are at a high, 25% risk of recurrence of this syndrome in subsequent pregnancies (autosomal recessive inheritance).

Summarizing, the clinical differentiation of Meckel–Gruber syndrome and Joubert syndrome in the prenatal period is not easy, but is pivotal for proper prognosis of survival, and high caution and accuracy in the interpretation of the results of genetic tests are required.

## Figures and Tables

**Figure 1 genes-12-01078-f001:**
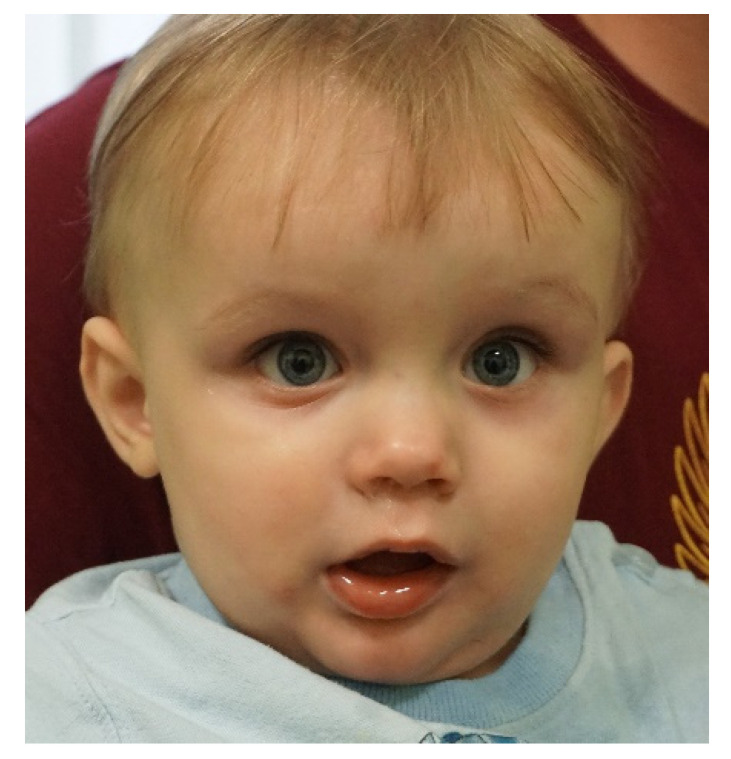
The child with Joubert syndrome.

**Figure 2 genes-12-01078-f002:**
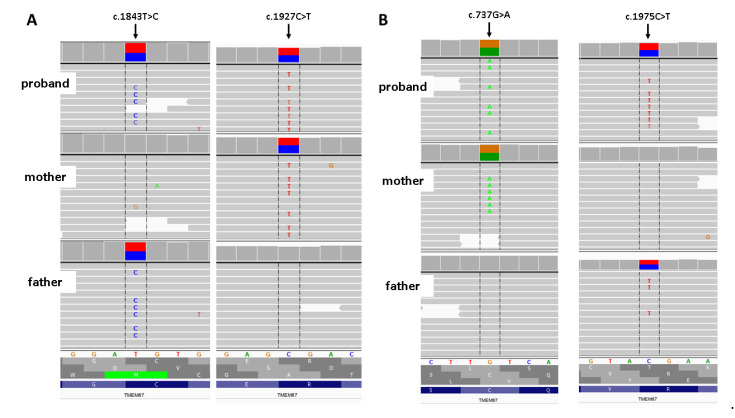
Results of family study. (**A**) Rresults of amplicon deep sequencing of case # 1. (**B**) Results of amplicon deep Integrative Genomic Viewer screenshot.

**Table 1 genes-12-01078-t001:** Characteristic of *TMEM67* variants identified in examined patients.

Patient.	#1		#2.	
Gene	*TMEM67* (NM_153704.5)			
Variant (GRCh38/hg38)	chr8:093795970-T > C c.1843T > C; p. (Cys615Arg)	chr8:093797200-C > T c.1927C > T; p. (Arg643Ter)	chr8:093780615-G > A c.737G > A; p. (Cys246Tyr)	chr8:093797345-C > T c.1975C > T; p. (Arg659Ter)
Parental origin	paternal	maternal	maternal	paternal
SNP association number (Rsid)	rs201893408	rs115195998	no applicable	rs150332116
Frequency ^a^	0.00009194	0.00001315	0.0	0.00003945
ACMG pathogenicity prediction ^b^	pathogenic	pathogenic	likely pathogenic	pathogenic
ClinVar ^c^	Pathogenic/likely pathogenic	no data	no data	no data

^a^ Frequency according to gnomAD database v3.1 (https://gnomad.broadinstitute.org/) accessed on 15 November 2020. ^b^ Pathogenicity prediction according to ACMG classification [8] (source https://varsome.com). ^c^ Reported in ClinVar database (https://www.ncbi.nlm.nih.gov/clinvar).

## Data Availability

Data is contained within the article.

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
