# Peer review of "Prenatal Versus Postnatal Diagnosis of Meckel–Gruber and Joubert Syndrome in Patients with TMEM67 Mutations"

_genes, 2021, doi:10.3390/genes12071078_

Round 1

Reviewer 1 Report

This paper reports the prenatal diagnosis of two cases with mutations in the TMEM67 transcript.

Specific comments

Since it is a recessive condition and the both cases were found to be compound heterozygotes, the sentence in the discussion :"We present clinical reports of two fetuses with two heterozygous mutations in the TMEM67 gene" should be modified for clarity

the sentence "These symptoms of the molar tooth sign changed the diagnosis from Meck-elGruber syndrome to Joubert syndrome." is not clear and should be modified in The presence of the molar tooth sign changed the siagnosis from Meckel-Gruber ......."

Author Response

Dear Reviewer #1

Thank you for all valuable comments, we tried to relate to each of them with due diligence. We hope that the responses will meet with your acceptation.

  1. English language and style are fine/minor spell check required

Answer: English language in the manuscript was checked by translation company, but again, according to wish of reviewer, the text of manuscript was checked by native speakers who is medical doctor. 

  1. Specific comments

- Since it is a recessive condition and the both cases were found to be compound heterozygotes, the sentence in the discussion: "We present clinical reports of two fetuses with two heterozygous mutations in the TMEM67 gene" should be modified for clarity –

Answer: It was corrected

- the sentence "These symptoms of the molar tooth sign changed the diagnosis from Meck-elGruber syndrome to Joubert syndrome." is not clear and should be modified in The presence of the molar tooth sign changed the diagnosis from Meckel-Gruber ......."

Answer: It was corrected

Reviewer 2 Report

This is a generally solid manuscript describing the prenatal presentation of two fetus with ciliopathies. It needs only moderate English improvement to be acceptable for publication.  However, I did not find the table or figure in order to evaluate the specific variants, and there are inadequate details in the methods. I would want to see these before I can fully evaluate scientific validity.

Abstract: 

1) "Congenital renal cysts can be classified..."  

2) Is MRI commonly used prenatally to diagnose anything? 

3) "Progress in molecular diagnostics and availability of exome sequencing procedures allows diagnosis of single-gene disorders..."

Clinical reports:

Case #1, paragraph #4: an electrophysiological examination of what??? AND non-standard spacing of Meck-elGruber syndrome

Case #2: "The family history was notable for the spontaneous loss of three of the mother's prior pregnancies in the first trimester"

Results of genetic tests: make the nomenclature standard between the two patients

Discussion:  Paragraph 3: Please define MKS, probably in the 3rd sentence.

Author Response

Dear Reviewer #2,

Thank you for all valuable comments, we tried to relate to each of them with due diligence. We hope that the responses will meet with your acceptation. We improved the results, the figure and table were added.

1.Moderate English changes required

Answer: English language in the manuscript was checked by translation company, but again, according to wish of reviewer, the text of manuscript was checked by native speakers who is medical doctor. 

2.Are the conclusions supported by the results?

Answer: Can be improved

3.This is a generally solid manuscript describing the prenatal presentation of two fetus with ciliopathies. It needs only moderate English improvement to be acceptable for publication.  However, I did not find the table or figure in order to evaluate the specific variants, and there are inadequate details in the methods. I would want to see these before I can fully evaluate scientific validity.

Answer: We are very sorry, but figure and table were not included by mistake. We are now including them in the hope that it will live up to expectations.

4.Abstract:

1) "Congenital renal cysts can be classified…"

Answer: Congenital renal cysts is a developmental disorders

2) Is MRI commonly used prenatally to diagnose anything?

Answer: The point is that these two tests are the most commonly used to detect defects, not that MR is the most commonly used.

The ultrasound remains the predominant modality for evaluating fetal defects, although fetal MRI has been increasingly used. The differences in using US or MRI depend on the center and clinical problem. MRI is used in bone and soft tissue congenital defects, also in kidney defects and even is severe oligo- or anhydramnios

3) "Progress in molecular diagnostics and availability of exome sequencing procedures allows diagnosis of single-gene disorders…"

Answer: It was changed

Clinical reports:

Case #1, paragraph #4: an electrophysiological examination of what???

Answer: of visual pathways, was added.

AND non-standard spacing of Meck-elGruber syndrome; 

Answer: it was corrected

Case #2: "The family history was notable for the spontaneous loss of three of the mother's prior pregnancies in the first trimester"

Answer: it was changed.

Results of genetic tests: make the nomenclature standard between the two patients

Answer: As suggested variant nomenclature between two patients was made, and we rewrite the sentence as follow:“In both cases, the WES study revealed a compound heterozygote in TMEM67 gene: NM_153704.5: c.[1843T>C];1927C>T]/p.(Cys615Arg);(Arg643Ter) in case #1 and NM_153704.5: c.[737G>A];[1975C>T]/p.(Cys246Tyr);(Arg659Ter) in case #2 (Figure 2) inherited from healthy parents.”

Discussion: Paragraph 3: Please define MKS, probably in the 3rd sentence.

Answer: The explanation was added.

Round 2

Reviewer 2 Report

Agnieszka and colleagues have now provided their figures and tables.  They have also made all edits requested.  I feel the manuscript is appropriate for publication.